# Ablation of Peroxiredoxin V Exacerbates Ischemia/Reperfusion-Induced Kidney Injury in Mice

**DOI:** 10.3390/antiox9080769

**Published:** 2020-08-18

**Authors:** Jiyoung Park, Eun Gyeong Lee, Ho Jin Yi, Nam Hee Kim, Sue Goo Rhee, Hyun Ae Woo

**Affiliations:** 1College of Pharmacy, Graduate School of Pharmaceutical Sciences, Ewha Womans University, Seoul 120-750, Korea; jypark89@ewhain.net (J.P.); skyever147@ewhain.net (E.G.L.); skagml929@ewhain.net (N.H.K.); 2College of Pharmacy, Graduate School of Applied Science and Technology for Skin Health and Aesthetics, Ewha Womans University, Seoul 120-750, Korea; yihoz@ewhain.net; 3Yonsei Biomedical Research Institute, Yonsei University College of Medicine, Seoul 120-752, Korea; rheesg@yuhs.ac; 4Biochemistryand Biophysics Center, NHLBI, National Institutes of Health, Bethesda, MD 20892, USA

**Keywords:** peroxiredoxin V, reactive oxygen species, renal ischemia/reperfusion, renal dysfunction

## Abstract

Ischemia/reperfusion (I/R) is one of the major causes of acute kidney injury (AKI) and associated with increased mortality and progression to chronic kidney injury (CKI). Molecular mechanisms underlying I/R injury involve the production and excessive accumulation of reactive oxygen species (ROS). Peroxiredoxin (Prx) V, a cysteine-dependent peroxidase, is located in the cytosol, mitochondria, and peroxisome and has an intensive ROS scavenging activity. Therefore, we focused on the role of Prx V during I/R-induced AKI using Prx V knockout (KO) mice. Ablation of Prx V augmented tubular damage, apoptosis, and declined renal function. Prx V deletion also showed higher susceptibility to I/R injury with increased markers for oxidative stress, ER stress, and inflammation in the kidney. Overall, these results demonstrate that Prx V protects the kidneys against I/R-induced injury.

## 1. Introduction

The kidneys are important organs of excretory systems, and these bean-shaped organs are about the size of one’s fist. Despite its small size, the kidney is highly vascularized, so the kidney receives high renal blood flow (RBF), about 20–25% of the cardiac output, 1000–1200 mL/min [1]. The waste products in the blood are filtered out into urine by glomerulus in which nephrons, the structural and functionally basic unit of the kidney, are located. The filtration function of the kidney is important for maintaining homeostasis. While, the prevalence of kidney disease has been increasing, but treatment of kidney failure is only limited to dialysis or transplant [2,3].

The kidney is highly sensitive to reactive oxygen species (ROS) [4,5,6]. The abnormal RBF induces oxidative stress through the generation of ROS. I/R is well recognized to cause RBF declines and stimulate ROS generation [7]. Along with chemotherapy-induced nephrotoxicity, ischemia/reperfusion (I/R) is one of the most causes of oxidative stress and is commonly used as an AKI. Ischemia initiates restriction of blood flow to the kidney which leads to hypoxia and expression of transcriptional genes such as hypoxia-inducible factor 1 (HIF1) and nuclear factor-kB (NF-kB). Subsequent reperfusion causes ROS generation through reoxygenation. This further induces inflammatory responses, such as cytokine (Interleukin-1β; IL-1β, Interleukin-6; IL-6, Interleukin-8; IL-8, tumor necrosis factor-α; TNF-α) and adhesion molecule adhering leukocyte (intercellular adhesion molecule; ICAM, vascular cell adhesion molecule; VCAM), infiltration of leukocyte, coagulation, and ROS generation. The generated ROS causes oxidative damage of lipid membrane, protein, DNA; cell death such as apoptosis, necrosis; microvascular dysfunction; and eventually kidney failure [8,9,10]. ROS includes superoxide, hydrogen peroxide, highly reactive hydroxyl radical, and is produced by UV light, ionizing radiation, an anti-cancer drug such as cisplatin, and renal vascular disease. The superoxide is reduced by superoxide dismutase (SOD). Hydrogen-peroxide-eliminating enzyme includes catalase, glutathione peroxidases (Gpxs), and Prxs in the enzymatic antioxidant system. Renal oxidative stress is developed from an imbalance between ROS production and antioxidant defense systems. Many antioxidant enzymes including Prxs maintain an appropriate level of ROS and prevent oxidative damages.

Prxs are a family of peroxidases that reduce peroxides and expressed in six forms in mammals (Prx I–VI) [11,12,13]. Prxs prevent oxidative damage in various tissue [14,15]. Among Prxs, Prx V eliminates not only peroxides but also peroxynitrites [16]. Recently, it was reported that Prx V inhibits adipogenesis by modulating ROS generation and adipogenic gene expression in vivo [17]. Prx V deletion in mice increased the susceptibility to high-fat diet-induced obesity and several of its associated metabolic disorders. Prx V contains two cysteine residues and thiol of the cysteine is oxidized to disulfenic acid and disulfide bond for peroxides reduction. Prx V gene of humans is located on chromosome 11q13 [18]. This gene contains two start codons; it is expressed as long-form Prx V (L-Prx V) containing mitochondria targeting sequence (MTS) and short-form Prx V (S-Prx V). MTS contains positively charged amino acids, so that L-Prx V is imported into inner mitochondria, which is negatively charged by pumping out of hydrogen ion through an electron transport system (ETS). S-Prx V contains a weak peroxisomal targeting sequence (PTS) at C-terminus and distributed in cytosol and peroxisome [19,20,21].

Prx V has been recently reported to prevent TGF-β-induced kidney injury through inhibition of signal transducer and activator of transcription 3 (STAT3) activation [22]. However, the role of Prx V in mediating the pathogenesis of AKI remains unclear. Thus, we hypothesize that Prx V is protective against I/R-induced kidney injury, which simulates oxidative stress.

## 2. Materials and Methods

### 2.1. Animals

Prx V WT and KO mice were generated by breeding of Prx V heterozygous mice [17]. Prx V WT and KO mice were identify by genotyping using polymerase chain reaction (PCR). Prx V mice were maintained on a 12/12 h (light/dark) cycle at 23 °C and supplied normal chow (LabDieat St. Louis, MO, USA) and water. All animal experiments were permitted by Ewha Womans University’s Institutional Animal Care and Use Committee (IACUC 17-059).

### 2.2. I/R Model

The mice were anesthetized with avertin (250 mg/kg, Sigma-Aldrich, St. Louis, MO, USA). Anesthetized mice were placed on a heating pad set at 37 °C, then shaved, perform a surgical site preparation of the incision site. Applying a 7.5% povidone solution with clean gauze in a circular fashion starting at the surgical incision site and rotating outward, then wiped with 70% ethanol. The renal pedicles were prepared by bilateral dorsal flank incision (1–1.5 cm) of the skin and muscle. Exposed two renal veins and arteries were tied using 3.0 non-absorbable suture (Ailee Co., Busan, Korea) at once to block blood flow to induce ischemia for 30 min. The color of the kidney was changed from red to purple. After color change, opened muscle and skin are sutured using a 3.0 non-absorbable suture. After 30 min, the suture that blocked the kidney’s vein and artery was removed to induce reperfusion [23]. Sham-operated mice underwent the same surgical procedure, except for the block blood flow to induce ischemia.

### 2.3. Subcellular Fractionation

After perfusion through the heart by phosphate-buffered saline (PBS, pH 7.4), kidneys were rinsed away any remaining blood and then collected kidneys were chopped and homogenized using a 7 mL Dounce homogenizer (Wheaton, NJ, USA) in 3 mL isolation buffer (10 mM Tris-HCl pH 7.4, 230 mM mannitol, 70 mM sucrose, 1 mM EDTA, 1 mM EGTA, 0.1% BSA) per a kidney on ice in order to break the cell membrane. The kidney homogenate was centrifuged at 1000× *g*, 4 °C for 10 min. The pellet which contained nuclei and the unbroken cell was removed. The supernatant was centrifuged at 12,000× *g*, 4 °C for 10 min. The resulting pellet contained mitochondria and peroxisomes and the supernatant contained cytosolic proteins. The resulting pellet was resuspended gently with 2 mL cold wash buffer (10 mM Tris-HCl pH 7.4, 230 mM mannitol, 70 mM sucrose) and centrifuged at 12,000× *g*, 4 °C for 10 min and repeated twice to wash [24,25].

### 2.4. Western Blotting

Kidneys were lysed with cold lysis buffer (20 mM HEPES pH 7.0, 0.15 M NaCl, 10% glycerol, 1% triton X-100, 1 mM EDTA, 1 mM EGTA, 10 mM β-phosphoglycerate, 1 mM NaVO_4_, 5 mM NaF, 1 μg/mL aprotinin, 1 μg/mL leupeptin, 100 μM PMSF) using homogenizer (Polytron, Brinkmann, Germany). The homogenates were centrifuged at 20,000× *g*, 4 °C for 15 min. After protein concentration of the lysates was quantified using Bradford assay (Bio-Rad, CA, USA), lysates were mixed with sample buffer (62.5 mM Tris-HCl pH 6.8, 10% glycerol, 2% sodium dodecyl sulfate (SDS), 0.0125% bromophenol blue, 2.5% β-mercaptoethanol) and boiled at 95 °C for 3 min. Samples were loaded onto a SDS-polyacrylamide gel electrophoresis gel and electrophoresed with SDS buffer (3 g/L Tris, 14.35 g/L glycine, 1 g/L SDS) to separate the proteins by size in the sample. The proteins were transferred onto the activated polyvinylidene difluoride (PVDF) membrane (Millipore, Darmstadt, Germany) with transfer buffer (3.03 g/L Tris, 14.17 g/L glycine, 20% methanol). The membrane was incubated with 5% skim milk in tris buffered saline with Tween-20 (TBST) at room temperature for 20 min using rocker to block nonspecific attachment of antibody on empty membrane between protein bands, followed by incubation at 4 °C for overnight using rocker with antibodies (1:2000 dilution). Anti-Prx I to VI [26,27], anti-GAPDH, anti-thioredoxin reductase (TR) [28], anti-thioredoxin (Trx) (Young In Frontier, Seoul, Korea), anti-sulfiredoxin (Srx) [26,27], anti-VDAC (Santa Cruz, CA, USA), anti-GPx I, anti-SOD (AbFrontier, Seoul, Korea) antibodies were used. The membrane was washed with TBST for 10 min each, in the high-speed rocker (FINEPCR, Gunpo, Korea). The membrane was incubated with horseradish peroxidase (HRP) conjugated-secondary antibodies (Bio-Rad, CA, USA) diluted 1:5000 and washed with TBST for 10 min each, in the high-speed rocker. Positive immunoreactive bands were detected with enhanced chemiluminescence (ECL) reagent (AbFrontier, Seoul, Korea) using the LAS-3000 (Fujifilm, Tokyo, Japan). The bands were quantified using the Multi Gauge 3.0 program (Fujifilm, Tokyo, Japan).

### 2.5. Histology

The isolated perfused kidney was fixed with 4% formaldehyde *w*/*v* at 4 °C for overnight, dehydrated, embedded in paraffin, cut into thickness of 4 μm using a microtome, and placed on a slide glass. The slide was incubated at 60 °C for 30 min, then deparaffinized with xylene, then hydrated with ethanol and distilled water [14,29].

Hematoxylin and eosin staining (H&E staining): The hydrated section on slide was immersed in mayer’s hematoxylin (Thermo Scientific, MA, USA) for 10 min and dipped twice in eosin Y (BBC Biochemical, VA, USA), then dehydrated with ethanol and xylene. Toluene was used for mounting.

Immunohistochemistry staining (IHC staining): A circular dam around the sections was made using a Dako-pen (Dako, Glostrup, Denmark), which is a hydrophobic pen. The slide was quenched endogenous peroxidase with blocking reagent (3% hydrogen peroxide, 97% methanol) for 15 min, then washed with PBS, then incubated with the blocking solution of the Impress Reagent Kit (Vector, CA, USA) at room temperature for 60 min, then incubated with 1:500 diluted anti-3-nitrotyrosine (3-NT) (Millipore, Darmstadt, Germany), anti-4-hydroxynonenal (4-HNE) (JalCA, Tokyo, Japan), anti-F4/80 (Abcam, Bristol, UK) antibodies at 4 °C for overnight. After washing with PBS, the slide was incubated with HRP-conjugated secondary antibodies at 4 °C for 150 min, then washed with PBS, then stained with 3,3′-diaminobenzidine (DAB, Vector, CA, USA) which was substrate of HRP and oxidized to brown color.

Terminal deoxynucleotidyl transferase dUTP nick end labeling (TUNEL) assay was performed on the paraffin sections with an in situ cell death detection kit (Sigma-Aldrich, St. Louis, MO, USA), according to the manufacturer’s instructions. They were examined with a confocal microscope (Nikon A1R, Tokyo, Japan). Images were processed using NIS-Elements software. Instruments (Nikon A1R) was supported by the Fluorescence Core Imaging Center on Ewha Womans University.

### 2.6. Measurements of Blood Parameters

Before sacrifice, blood samples were collected. The blood was collected from the inferior vena cava and the plasma was separated via centrifugation at 800× *g* for 15 min at 4 °C. Serum creatinine concentration (Scr) was measured with Creatinine Assay Kit (Bioassay System, CA, USA). Blood urea nitrogen (BUN) concentration was measured with an urea assay kit (Bioassay System, CA, USA).

### 2.7. RNA Isolation

The kidney was lysed with 1 mL TRIzol (Invitrogen, CA, USA) using an homogenizer on ice, then added 200 μL chloroform and vigorously vortexed for 15 sec and rested for 3 min in order to separate the phenol from lysate, then centrifuged at 12,000× *g*, 4 °C for 15 min. The lysate was separated into three layers. The aqueous-top layer contained RNA and was transferred to another tube. The tube was added 500 μL isopropanol and inverted gently four times and kept at −80 °C for 16 h for precipitation. Frozen mixture in the tube was melted on ice, then centrifuged at 12,000× *g*, 4 °C for 10 min. The pellet was washed twice with 70% ethanol in RNase free water and centrifuged at 12,000× *g*, 4 °C for 10 min, then dried. RNA pellet was resuspended with RNase free water. RNA purity and concentration were determined using Nano Drop ND-1000 spectrophotometer (Daemyung, Seoul, Korea).

### 2.8. Reverse Transcription-Polymerase Chain Reaction (RT-PCR) and Quantitative Real-Time PCR (qPCR)

The isolated RNA was subjected to RT-PCR in order to convert single-stranded RNA into stable double-stranded complementary DNA (cDNA). Two μg of RNA diluted with diethyl pyrocarbonate (DEPC) water (final volume 20 μL), then added to cDNA premix (EcoDry, CA, USA) which contained reverse transcriptase, deoxy-nucleotide (dNTP), random hexamer primers, and buffer. The tube was incubated at 42 °C for 60 min (reverse transcription) and then at 70 °C for 10 min (reverse transcriptase inactivation). The resulting cDNA was subjected to qPCR using ABI 7300 real time PCR system (Applied Biosystems, CA, USA). The reaction had 20 μL of a total volume, including 2 μL (40 ng) of cDNA, 10 μL of SYBR Green premix (Bioline, Bristol, UK), 0.25 pM of each forward and reverse primers, autoclaved DW. The primer sequences were listed in Table 1.

### 2.9. Statistical Analysis

The western blot protein bands were quantified via densitometry using ImageJ software (ImageJ 1.50I, Bethesda, MD, USA). All values were expressed as means ± standard error (SE). Statistical significance was analyzed via 2-factor ANOVA for multiple comparisons using the Graph Pad Prism software, version 6 (GraphPad, Bethesda, MD, USA). A *p*-value of <0.05 was considered statistically significant.

## 3. Results

### 3.1. IR-Induced AKI is Exacerbated by Ablation of Prx V

We first examined how Prx V protein changes during renal I/R. I/R was initiated by clipping the renal blood vessels for 30 min followed by release. Thereafter, the kidney damage was examined at 72 h after ischemia, which is the acute kidney injury stage. Prx V is remarkably widespread among subcellular compartments compared to other Prxs. To confirm this, mitochondrial and cytosolic fractionated from kidney cells and then were subjected to immunoblot analysis. There was no significant difference between whole and cytosolic Prx V level, and mitochondrial Prx V was significantly reduced in I/R injured mice (Figure 1A). The mRNA levels of mitochondrial Prx V (L-Prx V) and total Prx V were not changed (Figure 1B).

Next, we investigated the kidney function of Prx V WT and KO mice, to confirm the effects of Prx V on I/R injury. The kidneys were subjected to H&E staining to observe the structural change of the kidney. As a result, loss of brush border on the proximal tubule and formation of the cast in the tubule was more severe in Prx V KO mice than Prx V WT (Figure 2A). The cast leads to stenosis and dilation in the tubule, and renal dysfunction. We also evaluated ischemic bodyweight. As represented in Figure 2B, the ischemic bodyweight of Prx V KO was more decreased compared to Prx V WT after I/R induced acute injury. BUN and Scr that are the byproducts of metabolism of protein and muscle in the body and filtered out from the kidney are used to evaluate the kidney function. In particular, creatinine is filtered but not reabsorbed as regards the index of glomerular filtration. The serum level of BUN and Scr indicated AKI was significantly enhanced in Prx V KO mice compared with Prx V WT mice (Figure 2C). Ischemic damage markers showed no significant difference between Prx V WT and KO sham-operated mice. Prx I and II are also well recognized ROS scavenger in vivo [14,30], so we tried I/R injury to Prx I or II KO mice to confirm their effects. To assess whether Prx I or II KO mice exhibit the kinetics of renal function decline, BUN and Scr were measured. There was no difference in renal function decline between WT and KO mice (Figure 2D,E).

In order to confirm the change of antioxidant proteins during I/R, the kidneys lysates were subjected to immunoblot analysis. The levels of antioxidant proteins Prxs, Gpx I, TR, SOD, and Trx appeared to be not significantly different between the sham-operated and I/R group both in Prx V WT and KO mice (Figure 3A). Immunoblots confirmed that Srx was increased in kidneys from the I/R group (Figure 3A). Srx that is an oxidative stress induced-protein as the target gene of Nrf2 [31,32] was increased during I/R injury. Prx V KO mice have significantly increased Srx induction compared to Prx V WT during I/R (Figure 3A,B).

From these results, we could hypothesize that Prx V is an important antioxidant enzyme for kidney injury.

### 3.2. Ablation of Prx V Induces More Inflammatory Responses by Renal I/R Injury

Excessive inflammation is commonly found in AKI [33,34]. To confirm whether there is any difference in the infiltration of immune cells, we stained the sections for macrophages (F4/80). The macrophage infiltration was increased in I/R induced kidney and was highly enhanced in KO mice compared with Prx V WT (Figure 4A). The mRNA levels of pro-inflammatory mediators, i.e., TNF-α and IL-1β were highly elevated in Prx V KO mice compared to Prx V WT mice during I/R (Figure 4B). Altogether, the inflammatory response due to in I/R injury was exaggerated in Prx V KO mice.

### 3.3. Ablation of Prx V Induces Highly Enhanced Oxidative Stress, ER Stress, and Apoptosis by Renal I/R Injury

Ischemia induced-hypoxia leads to Nrf2 expression and Srx expression as downstream of Nrf2. mRNA level of Nrf2 and Srx in the kidney of Prx V KO was increased compared to that in the kidney of Prx V WT during I/R (Figure 4B). Oxidative stress in sham and I/R-indueced kidneys was assessed by staining for 3-NT, a marker of peroxynitrite formation, and 4-HNE, a marker of lipid peroxidation. Immunohistochemical analyses presented that the expression of 4-HNE and 3-NT adducts in the centrilobular areas of the kidneys were increased by I/R induced injury and were increased further by I/R induced in Prx V KO mice (Figure 5A,B). Oxidative stress was exaggerated in I/R-induced Prx V KO mice presented by an increased IHC-stained kidney section. Oxidative stress in AKI induces ER stress and apoptosis [35,36]. Furthermore, mRNA abundance of the ER stress markers, i.e., ER degradation-enhancing a-mannosidase-like protein 1 (Edem), Tribbles homolog 3 (TRB 3), x box-binding protein 1 (XBP1s), and AMP-dependent transcription factor (ATF 4) were significantly elevated in I/R induced mice, ER stress was enhanced by ablation of Prx V (Figure 4B). Quantification of TUNEL-positive cells, representing internucleosomal DNA fragmentation, showed more TUNEL-positive cells in Prx V KO than Prx V WT mice (Figure 5C). These results suggest that Prx V serves to protect acute kidney injury from apoptosis caused by I/R.

## 4. Discussion

There is accumulating evidence that suggests that an important consequence of I/R injury at least partly contributes to the high morbidity and mortality rates of patients with AKI [37]. A number of conditions, including kidney transplantation, induce renal I/R injury; this causes problems for many recipients of kidneys and may therefore negatively impact postoperative consequences [38]. Oxidative stress, induced by the pathological overproduction of ROS and reactive nitrogen species, plays a substantial role in the development of renal I/R injury [39]. Ischemia leads to hypoxia by restriction of blood flow supply, lack of oxygen causes change of aerobic metabolism to anaerobic metabolism for cell survival, but subsequently reperfusion leads to ROS generation by rapid restoration of blood. The outburst of highly electrophilic ROS in the reperfusion process perturbs the balance of renal redox state, which directly causes renal tubular cell damage functionally and structurally by extensive membrane lipid peroxidation, DNA breakdown, and protein inactivation [40]. Furthermore, the excessive ROS attacks the cells, leads to immune responses, cell apoptosis. Many antioxidant enzymes including Prxs maintain an appropriate level of ROS and prevent oxidative damage. Prx V is ubiquitously expressed in many subcellular compartments [41,42,43] considered as a cytoprotective antioxidant without being inactivated by hyperoxidation [16,44,45]. Prx V has an intensive ROS scavenging activity and has a unique activity for peroxides and peroxynitrites [16].

To demonstrate the role of Prx V in I/R-indueced AKI in vivo, we employed a KO mouse. This I/R model explained the protective effect of Prx V in correlation to ROS and AKI. In the present study, mitochondrial Prx V (L-Prx V) decreased in protein level without accompanied by changes in mRNA level in response to I/R operation (Figure 1). Prx V KO kidneys showed excessive body weight loss and decline in renal function during I/R when compared to the I/R-Prx V WT mice (Figure 2). These phenotypic changes support that Prx V mediates AKI. Srx protein as an oxidative stress marker was more increased in I/R-Prx V KO mice (Figure 3). Immunohistochemical analyses presented that the expression of 4-HNE and 3-NT adducts were increased further by I/R induced in Prx V KO mice (Figure 5). These results indicated that ablation of Prx V exacerbated oxidative stress. Prx V deletion also aggravated I/R injury, supported by increased markers of inflammation, ER stress, and apoptosis in mouse kidney (Figure 4 and Figure 5).

## 5. Conclusions

Our results indicate that Prx V plays a protective role in I/R induced-kidney injury and that it could be a potential therapeutic target for AKI or chronic kidney disease.

## Figures and Tables

**Figure 1 antioxidants-09-00769-f001:**
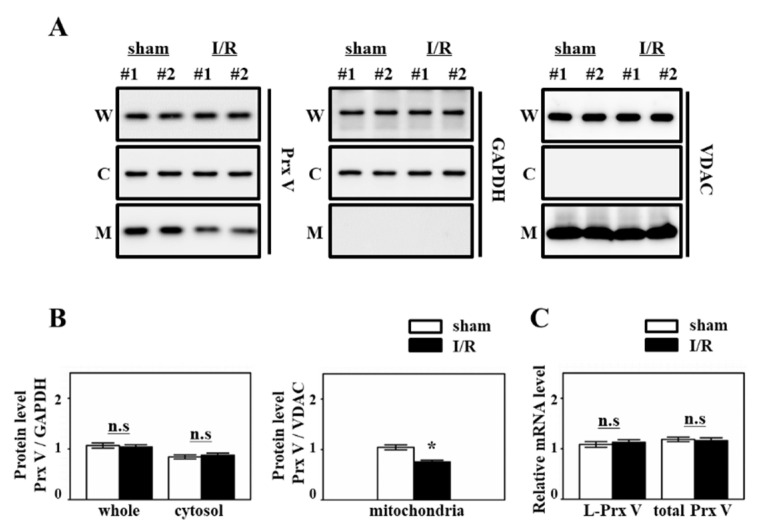
Prx V expression in mice during renal I/R. (**A**) Protein expression of Prx V was analyzed by immunoblotting. Kidneys were subjected to subcellular fractionation. W: whole lysates, C: cytosol fraction, M: mitochondrial fraction. (**B**) Prx V protein level of (**A**) was quantified by Multi Gauge V3.0 software (Fujifilm, Tokyo, Japan) and normalized to GAPDH as control of cytosolic protein or VDAC as control of mitochondrial protein. (**C**) The mRNA expression level of mitochondrial Prx V (L-Prx V) and total Prx V in I/R induced kidney. Data are shown as means ± SE. *n* = 4–6/group * *p* < 0.05 vs. sham, not significant; n.s.

**Figure 2 antioxidants-09-00769-f002:**
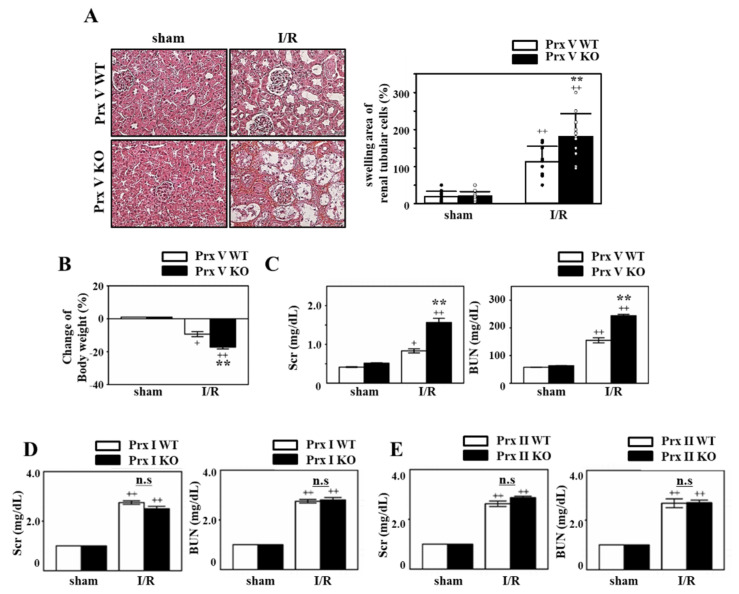
Effect of Prx V ablation on renal structure and function during renal I/R. Prx V WT and KO male mice were subjected to renal ischemia for 30 min followed by up to three days of reperfusion or sham surgery. (**A**) The kidney sections were subjected to H&E staining. Original magnification, 400×. Quantification of the swelling area of the renal tubular cells. NIS-Elements AR 3.1 software was used to quantify. (**B**) Changes of body weight were measured before sacrificed. (**C**) BUN and Scr concentrations were measured from serum of the mice. Prx I (**D**) or II (**E**) WT or KO mice were subjected to renal ischemia for 30 min followed by up to 3 days of reperfusion or sham surgery. BUN and Scr concentrations were measured from serum of the mice. Data are shown as means ± SE. *n* = 5–6/group, ** *p* < 0.01 vs. WT, + *p* < 0.05 vs. sham, ++ *p* < 0.01 vs. sham, not significant; n.s.

**Figure 3 antioxidants-09-00769-f003:**
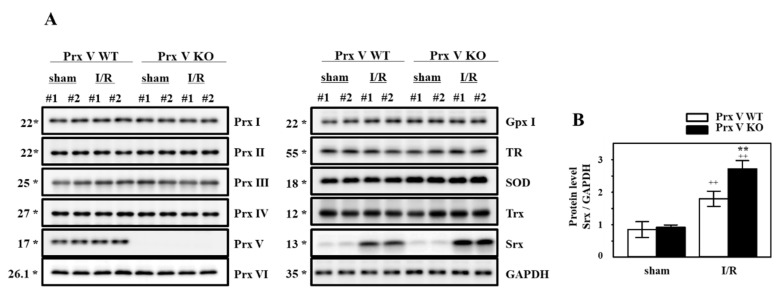
Effect of Prx V ablation on antioxidant proteins during renal I/R. (**A**) The expression of antioxidant proteins was analyzed by immunoblotting. (**B**) Srx protein level in (**A**) was quantified by Multi Gauge V3.0 software and normalized to GAPDH. Data are shown as means ± SE. *n* = 4–6/group * *p* < 0.05 vs. sham, ** *p* < 0.01 vs. sham, ++ *p* < 0.01 vs. sham.

**Figure 4 antioxidants-09-00769-f004:**
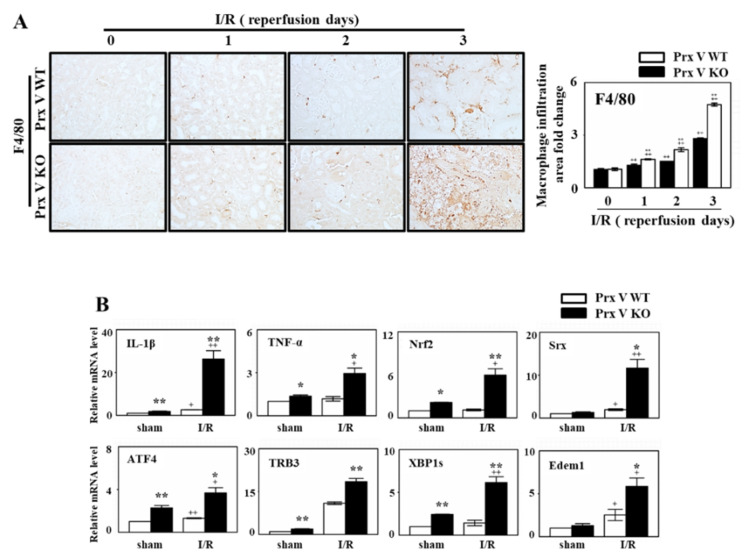
Effect of Prx V ablation on macrophage infiltration and oxidative stress-related mRNA levels during renal I/R. (**A**) The kidney sections were subjected to IHC with antibodies to F4/80 as macrophage marker. NIS-Elements AR 3.1 software was used to quantify. Graph is relative macrophage (F4/80) infiltration area fold change. Original magnification, 400× (**B**) The mRNA expression level of oxidative stress, ER stress and apoptosis-related genes in the mice kidney. Data are shown as means ± SE. *n* = 5–6 * *p* < 0.05 vs. WT, ** *p* < 0.01 vs. WT, + *p* < 0.05 vs. sham, ++ *p* < 0.01 vs. sham.

**Figure 5 antioxidants-09-00769-f005:**
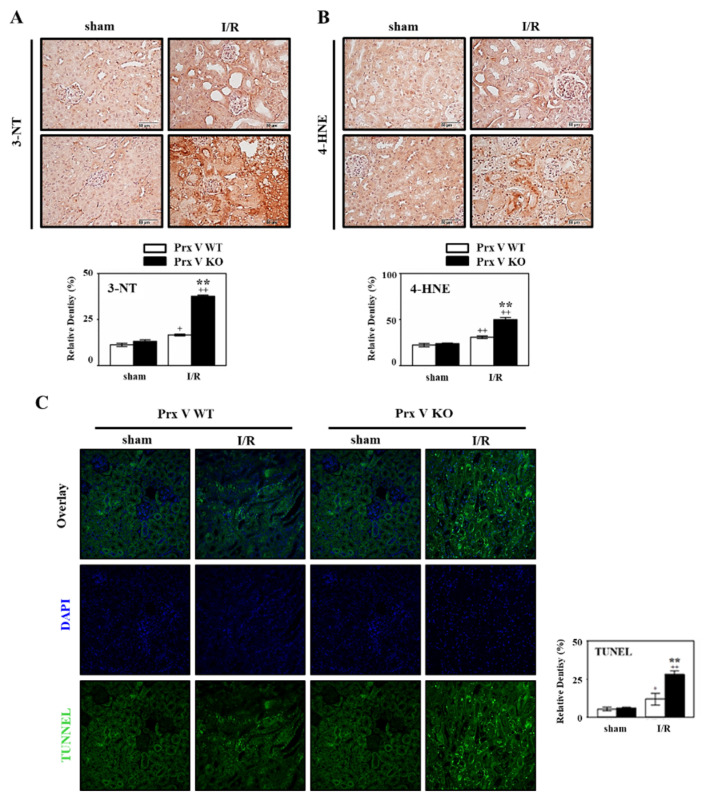
Effect of Prx V ablation on oxidative damage and apoptotic damage during renal I/R. (**A**) IHC with antibodies to (**A**) 3-NT and (**B**) 4-HNE as oxidation marker of tyrosine and lipid. (**C**) The kidney sections were subjected to TUNEL assay. NIS-Elements AR 3.1 software was used to quantify. Original magnification, 400×. Data are shown as means ± SE. *n* = 5–6, ** *p* < 0.01 vs. WT, + *p* < 0.05 vs. sham, ++ *p* < 0.01 vs. sham.

**Table 1 antioxidants-09-00769-t001:** Sequence of quantitative real-time polymerase chain reaction (qPCR) primers.

Target Gene	Forward Primer (5′-3′)	Reverse Primer (5′-3′)	Size
GADPH	AGAACATCATCCCTGCATCC	GGTCCTCAGTGTAGCCCAAG	228
LPrx V	AGAAGCAGGTTGGGAGTGTG	CTTTCTTGCCCTTGAACAGC	158
SPrx V	GGCATTTACACCTGGCTGTT	CGACGATTCCCAAAGAGAGA	242
Nrf-2	CTCTCTGAACTCCTGGACGG	GGGTCTCCGTAAATGGAAG	182
Srx	GGAAGGAAGAAAGGAGATGG	AGAGTTCAGGCTATGGGGAT	155
ATF4	ATGGCCGGCTATGGATGAT	CGAAGTCAAACTCTTTCAGATCCATT	113
TRB3	CTCTGAGGCTCCAGGACAAG	GGCTCAGGCTCATCTCTCAC	142
XBP1s	GAGTCCGCAGCAGGTG	GTGTCAGAGTCCATGGGA	149
Edem 1	GCAATGAAGGAGAAGGAGACCC	TAGAAGGCGTGTAGGCAGATGG	157
IL-1β	TCGTGCTGTCGGACCCATAT	GTCGTTGCTTGGTTCTCCTTGT	110
TNF-α	GCCACCACGCTCTTCTG	GGTGTGGGTGAGGAGCA	294

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
