# Peer review of "Ablation of Peroxiredoxin V Exacerbates Ischemia/Reperfusion-Induced Kidney Injury in Mice"

_antioxidants, 2020, doi:10.3390/antiox9080769_

Round 1
Reviewer 1 Report
In this study, the authors explored the in vivo role of peroxiredoxin V (Prx V) in ischemia/reperfusion (I/R) induced-kidney injury. For this purpose, the authors created knockout mice, in which the Prx V gene was functionally inactivated, and compared these knockout animals with outbred wildtype controls. They found that inactivation of the Prx V gene enhanced the susceptibility of mice against to I/R-induced oxidative stress. In fact, Prx V KO mice exhibited more severe renal damage than wildtype mice in this particular I/R injury model. From these data the authors concluded that Prx V is protective against I/R-induced kidney injury. This is an interesting study looking into the biological role of Prx V in the kidney. The experiments were well planned and have been carefully executed. Despite my overall positive impression, I have three major concerns and a number of more specific critical points, which should be addressed by the authors in case the handling editor invites resubmission of a suitably revised version of the ms.
Major concerns
- The authors neither describe in the ms how the Prx V knockout mice were generated nor where these animals come from. In the Materials and methods section (2.1.) the authors report that Prx heterogeneous (what exactly is heterogeneous or do the authors mean heterozygous) males and females were mated to produce Prx V KO and WT mice. Unfortunately, it remains unclear how the heterozygous Prx KO mice were created. What kind of knockout strategy was employed (general stem cell KO, tissue specific conditional KO)? How exactly was the Prx V gene functionally inactivated (induction of a neomycin resistance cassette, frame shift mutation or other methods)? How was genotyping performed to separate WT from KO mice after crossbreeding of the heterozygous animals?
- Have the homozygous Prx V KO mice been characterized before? Do they have a resting phenotype? The Prx V KO mice are obviously viable but are there any phenotypic alterations? Do they breed well or is reproduction impaired by the gene knockout? Have the body weight kinetics measured and is the aging process impacted? Are there any hematopoietic defects in the Prx V KO mice? If the phenotype of the KO mice has not been characterized before such data should be given in this ms. If such data have been published before the corresponding papers should be referenced. If functional characterization of these mice is the topic of a separate paper some phenotypic essentials should at least be verbally described in this ms. Without basic functional characterization of these mice the present study may not be acceptable for publication.
- The study is very descriptive and lacks mechanistic focus. The authors described structural and functional alterations they observed in Prx V KO mice but they did not explore the molecular mechanism the mechanistic links between the different readout parameters. For instance, the authors found (Fig. 4) that the mRNA concentrations of IL1ß, TNFa, Srx and others are higher in Prx V KO mice than in WT animals but it remains unclear this is the case. How exactly are Prx V and IL1ß mechanistically interconnected.
Specific points of criticisms
a) Fig. 1: This figure shows that the I/R model the authors employed induces structural and functional changes in WT mice. Since this model has been described before the image simply confirms that the model did work in the hands of the authors. This conclusion can be stated in the text and does not need a separate figure. Moreover, the data reported in Fig. 1 are also shown in Fig. 3 and thus, Fig. 1 and Fig 3 are in part redundant. In fact, Fig. 1 should be omitted.
b) Fig. 2: For most readout parameters the authors did not find any difference between the sham and the I/R treated animals. Thus, this image presents mostly negative data. These findings can simply be described in the text and no extra figure is needed.
c) Fig. 2: VDAC was detected in the mitochondrial fraction but not in the whole cell lysate. Since the mitochondria should be present in the whole cell lysates it is difficult to understand why there is no VDAC signal in the whole cell lysate.
d) Which GPx isoform was quantified and which antibody was used?
In the light of these comments I feel myself unable to recommend the ms in its present form for publication in Antioxidants. However, id the authors feel confident to address the critical points raised in the editorial report resubmission of a revised version of the paper should be invited.
Reviewer 2 Report
The manuscript provides important, new insight into Prx V and the impact knockout of the encoding gene has on ischemia/reperfusion-induced kidney injury.
All of the comments listed below could likely be addressed through rewording the manuscript.
The manuscript title is misleading. The authors demonstrate that the absence of peroxiredoxin V (PrxV) renders mice sensitive to ischemia/reperfusion-induced kidney injury. Note that the absence of PrxV could impact other pathways in the mice. Evidence is not presented to support a protective role of PrxV against ischemia/reperfusion-induced kidney injury in mice. The latter would require overexpression of PrxV.
Srx appears to undergo a dramatic increase in protein abundance during I/R (30 min). However, the authors provide little if any information and/or literature citations to define where Srx located in the mouse and whether Srx knockout mice are sensitive to I/R-induced kidney injury.
Ln 19 Prx V is highly expressed in kidney and decreased by – unclear whether this phrase is describing transcriptional expression rates, transcript levels or protein abundance
Ln 23 the phrase [were worse] is subjective – please provide a quantitative adjective to describe the changes that occurred
Please have someone correct the grammar issues that are prevalent throughout the entire manuscript. Nearly every sentence has some type of grammatical error. For example,
Ln 13 flow in kidney of mice – flow in the kidneys of mice
Ln 14 as acute and chronic kidney injury model – as an acute and chronic kidney injury model
Please define all abbreviations upon first use (e.g., Ln 47 AKI). After reading the manuscript, the table of abbreviations was found. Recommend referring the reader to this table at an appropriate stage in the introduction.
Be sure to remove the template info such as follows:
“The introduction should briefly place the study in a broad context and highlight why it is 29 important. It should define the purpose of the work and its significance. The current state of the 30 research field should be reviewed carefully and key publications cited. Please highlight controversial 31 and diverging hypotheses when necessary. Finally, briefly mention the main aim of the work and 32 highlight the principal conclusions. As far as possible, please keep the introduction comprehensible 33 to scientists outside your particular field of research. References should be numbered in order of 34 appearance and indicated by a numeral or numerals in square brackets, e.g., [1] or [2,3], or [4–6]. See 35 the end of the document for further details on references.”
Make sure the paragraphs are organized in a logical manner. For example, ln 68-69 [Among Prxs, Prx I and II are abundant in cytosol in kidney tissue and Prx III is located only mitochondria.] is the first sentence of the last paragraph of the introduction but is not the main point of the manuscript. Need to rethink the topic sentence of this paragraph.
Be sure to use proper nomenclature – e.g.., ln 70 TGF-b should be TGF-beta (likewise later use of TGF-a should be TGF-alpha)
Please verify the accuracy of all references - for example, Ref 39 seems to have a glitch with two different M00 numbers. Furthermore, make sure to follow the reference format for Antioxidants. For example, the standard abbreviations should be used for the journal names.
Ln 75 please define the composition and quantity of normal chow
Ln 81 please define the quantity of avertin
Ln 82, what is the % povidone and is this mixed with the 70% ethanol or a separate wipe
Ln 90, define the stage at which the kidney is isolated.
Ln 94, x g is a universal abbreviation; thus, clarifying it as relative centrifugal force (RCF) is not necessary
Ln 90-99, what controls were used to ensure the stated fractionations were not cross contaminated (e.g., mitochondria were not in the cytosolic fraction, etc.)
Ln 99, what controls were used to ensure that the BSA was fully removed by the washing methods stated.
Ln 105 and 146, 15,000 rpm and 3,000 rpm change to x g
Ln 104, assume homogenization was similarly to the previous section; however, please provide details on the sonication
Ln 172, what controls were used to ensure removal of DNA
Ln 108, cooked at – recommend incubated at – since cooking implies that the samples will be eaten.
Ln 114, details on the use of the rocker (type, speed).
Ln 118, seem to be missing wash steps.
Ln 124, 4% formalin define % as w/v or v/v for this and other chemical formulations
Ln 124-127, not written with enough detail to ensure reproducibility. Is there some type of methods paper that can be referenced?
Ln 129, uppercase lettering for each word of company names - throughout manuscript
Ln 141, according to the manufacturer’s instructions - please define the manufacturer under discussion
Ln 145, [The blood was collected from the inferior vena cava] please provide more details on the sample collection such as timing.
Table 1. please include anticipated size of qRT-PCR product
Ln 196, please define BUN and Scr in ln 193
Figure 1A requires higher resolution, as the images and the lettering are difficult to visualize even in zoom mode. Please define the sample names.
Ln 191-192, [As a result, loss of brush border on the proximal tubule and formation of cast in the tubule was increased in kidney of I/R mice] please use the images to guide the reader on this point. In addition, clarify the level of increase in a quantitative manner.
Ln 197, Define what is meant by sham
Ln 198, define what is meant by the term “downstream” in terms of mechanism
Ln 200, the statement seems to be referring to Figure 2A, yet is cited as ref. [25, 26].
Figure 2A, please indicate the molecular mass markers on the left of each blot to provide perspective on the kDa of each protein band detected by western blotting analysis. In addition, please provide the negative control to ensure that each protein band is according to what is indicated by the authors. The biological authenticity of each of the antibodies used to detect a particular target protein needs to be controlled to ensure specificity under the conditions used for this study.
Ln 202-204, [Protein level of only mitochondrial PrxV was significantly decreased because of degradation not reduction of protein expression from mRNA]. The differences observed in PrxV protein abundance appear modest. While the data are shown as means ± SE (ln 230), the sample number and type (e.g., biological, technical, experimental replicates) are not provided making it difficult to discern the significance of this claim. Translation rates can impact protein abundance; thus, it is premature to claim that protein degradation is responsible for any differences observed in PrxV protein vs. mRNA levels.
For each figure legend, please define sample number and type (e.g., biological, technical, experimental replicates). For each blot, provide markers to indicate where the molecular mass standards migrated in the region cropped and presented.
Ln 187, Throughout this section –please clarify what is new insight and what is comparable to what has already been observed in the literature. For example, the authors state that [Prx I and II also is located in cytosol like Prx V]. Are these findings novel to this particular study?
Fig. 3. As a control, can the phenotype observed for the Prx V knockout mice be restored to wild type by ectopic expression of Prx V?
Ln 207-209. Please use Figure 3 panel letters A to E to guide the reader through the following statement [The Prx V KO mice was significantly observed tubular cell destruction and cast and tubular dilation in kidney and body weight loss and renal failure and Srx induction compared to Prx V WT]. In addition, please clarify what features were used to quantify the findings presented in panel A.
Figure 4A and 4B are presented out of order.
Ln 245, Unclear what is meant by the statement that [inflammation related-gene was determined.]
Figure 4A right panel - please define the y-axis units.
Ln 270, [It was confirmed through] reword - since the findings of this assay appear to provide new insight instead of confirming prior study.
Round 2
Reviewer 1 Report
The authors addressed in their rebuttal letter satisfactorily most of the critical comments I raised during the first round of evaluation. They clarified origin of the genetically modified mice and extensively restructured the ms. Unfortunately, the ms is still very descriptive and lacks major mechanistic focus. The authors describe that peroxiredoxin V is protective against ischemia/reperfusion-induced kidney injury but neither of the experiments provides a plausible explanation for this finding on the molecular level. In the rebuttal letter the authors speculate that that Prx V may trigger NFkB signaling but they did not show experimentally that this is really the case and the possible molecular mechanisms remain elusive. It is correct that investigations into the molecular basis of the observed effects are laborious and time consuming but in the absence of such data the study remains purely descriptive.
The basic question that remains to be answered is, whether the journal accepts such descriptive studies for publication. If so, the ms in its revised version can be accepted for publication as it is. If not, the ms should be rejected since I do not think that an additional round of revision will provide such missing mechanistic links. A principal decision of the handling editor is now needed.